# Design and Feasibility Verification of Novel AC/DC Hybrid Microgrid Structures

**DOI:** 10.3390/s24154778

**Published:** 2024-07-23

**Authors:** Jiaxuan Ren, Shaorong Wang, Xinchen Wang

**Affiliations:** School of Electrical and Electronic Engineering, Huazhong University of Science and Technology, Wuhan 430074, China; m202271906@hust.edu.cn (J.R.); m202271896@hust.edu.cn (X.W.)

**Keywords:** AC/DC hybrid microgrid, distributed energy, thyristor switch, droop control, power quality

## Abstract

To enhance the power supply reliability of the microgrid cluster consisting of AC/DC hybrid microgrids, this paper proposes an innovative structure that enables backup power to be accessed quickly in the event of power source failure. The structure leverages the quick response characteristics of thyristor switches, effectively reducing the power outage time. The corresponding control strategy is introduced in detail in this paper. Furthermore, taking practical considerations into account, two types of AC/DC hybrid microgrid structures are designed for grid-connected and islanded states. These microgrids exhibit strong distributed energy consumption capabilities, simple control strategies, and high power quality. Additionally, the aforementioned structures are constructed within the MATLAB/Simulink R2023a simulation software. Their feasibility is verified, and comparisons with the existing studies are conducted using specific examples. Finally, the cost and efficiency of the application of this study are discussed. Both the above results and analysis indicate that the structures proposed in this paper can reduce costs, improve efficiency, and enhance power supply stability.

## 1. Introduction

Constructing a power system focused on renewable energy is crucial for energy conservation and emission reduction [1]. In recent years, the share of renewable energy in overall energy consumption has increased [2], with distributed energy increasingly becoming the primary form of renewable energy. Microgrids are key to utilizing distributed energy and can effectively integrate them [3,4]. They can operate independently or interact with external grids, offering broad application prospects [5,6]. In islanded mode, microgrids mainly rely on internally distributed energy for power supply [7]. In grid-connected mode, the external grid can support the microgrid, and excess energy generated by distributed sources can be transferred to the external grid [8], thus balancing energy flows and avoiding energy waste, such as the curtailment of wind and photovoltaic power [9,10].

Microgrids can be classified into DC, AC, and AC/DC hybrid types based on their configuration [11]. Compared with AC microgrids, DC microgrids reduce the need for inverter processes due to the numerous distributed power sources (e.g., photovoltaics and batteries) that generate DC power [12,13]. Additionally, DC microgrids avoid issues including synchronization [14], power quality [15] and reactive power compensation [16]. However, due to the widespread presence of AC loads and sources in current grids, recurrent AC/DC conversions can lead to considerable energy losses, and a full transition towards DC grid structures from AC poses significant financial burdens. Consequently, AC/DC hybrid microgrids have emerged as a viable solution, integrating the advantages of both AC and DC microgrids.

AC/DC hybrid microgrids have many advantages: they promote energy management and improve the power quality of the system [17,18], integrate new power generation points or consumer points [19], do not need to synchronize additional ESS and PV by directly connecting to the DC bus [19], significantly reduce voltage transformation [20], and a reduction in converters reduces power loss, ensuring the high economy of the structure [21]. By integrating AC and DC microgrids, this structure can achieve higher reliability and stability [22]. The AC/DC hybrid microgrid, with its aforementioned advantages, represents the mainstream direction of microgrid development today. However, compared to traditional microgrids, it still exhibits weaknesses in terms of control complexity and relay protection complexity. Furthermore, the analysis of power flow is more intricate than in traditional power systems [23]. The transitions between multiple operational modes need to be smooth, and stability during operation also requires assurance. Additionally, adaptive protection against various faults must be implemented [24]. These challenges necessitate further research to develop more adaptable AC/DC hybrid microgrid systems.

Currently, various typical network topologies for both AC and DC microgrids, including radial, ring-shaped, and mesh structures, have been developed, with their respective advantages and disadvantages well known [25,26]. However, AC/DC hybrid microgrids are still in the developmental stage, and the construction of an economical and reasonable topology remains to be explored. The authors previously [27,28] designed and validated two types of AC/DC hybrid microgrid topologies suitable for residential and commercial applications, respectively. Moreover, AC/DC hybrid microgrids face several challenges, such as stability, relay protection, and coordinated control [29]. To address them, [30] proposed utilizing a multi-intelligent body system approach to improve the response speed and decision-making accuracy of the system, thereby enhancing voltage stability within the microgrid. The authors in [31] proposed a simplified model of the AC/DC hybrid microgrid for fault analysis through the equivalent simplification method of a mathematical model. In previous studies [32,33], the authors conducted analysis and studies on the coordinated control of energy storage modules, converters, and other components within AC/DC hybrid microgrids.

In recent years, significant progress has been made in the study of AC/DC hybrid microgrids and their cluster structures, particularly in the areas of coordinated control and optimized operation [34,35]. However, the issue of power supply reliability, especially during power source failures, remains a critical challenge that has not been explored in depth. To address it, the main research objectives of this paper are as follows: Firstly, to propose a novel AC/DC hybrid microgrid cluster structure capable of swiftly restoring power supply with minimal transition time in the event of a power source failure; secondly, to design islanded and grid-connected microgrids, respectively, according to different external power supply environments; and thirdly, to validate the proposed structure through time-domain simulation and compare the performance with existing studies. The rest of this paper is organized as follows: In Section 2, the novel AC/DC hybrid microgrid cluster structure is designed. In Section 3, the grid-connected and islanded AC/DC hybrid microgrid structures are designed. In Section 4, the control strategy for fast access to the backup power is introduced in detail. In Section 5, two evaluation indicators for assessing microgrid performance are introduced. In Section 6, the feasibility of the proposed microgrid structures is verified by simulation in MATLAB/Simulink, and the performance is compared with existing studies. In Section 7, the conclusions and limitations of this work are given.

The main innovations and contributions of the study in this paper are listed as follows:The novel AC/DC hybrid microgrid cluster structure proposed in this paper leverages the rapid response characteristics of thyristor switches. This enables the structure to effectively reduce the power outage time and swiftly restore power supply in the event of a power failure, thereby enhancing the reliability of power supply as well as the security and stability of the power system.The islanded and grid-connected microgrid structures designed in this paper efficiently harness distributed energy. The control strategies of the structures are simple, and the power quality is high. Furthermore, the structures are capable of adapting to changes in load operation modes, exhibiting high stability, economy, and practical application value.

## 2. A Novel AC/DC Hybrid Microgrid Cluster Structure

In this paper, a novel topology structure for the AC/DC hybrid microgrid cluster is proposed, as shown in Figure 1. The structure adopts a sectionalized single-bus configuration and is powered by two power sources. One of these power sources serves as the supply source, while the other acts as the backup source. The voltage from the 110 kV/35 kV power sources is stepped down, respectively, by transformer T1 and transformer T2 and then connected to the 10 kV bus. In the event of a failure of the supply source connected to the 10 kV bus, the thyristor switch will be triggered to switch to the other section of the bus. The switching process is brief, and the reliability of the power supply is high. The control strategy followed by this process will be described in detail in Section 4. Conventional loads are connected to both sections of the 10 kV bus and can be energized or de-energized according to the actual situation. During normal operation, the thyristor switch is off, while the disconnecting switch K1 or K2 and the circuit breaker B1 are closed to supply power to the critical loads. Furthermore, several microgrids and energy storage batteries are connected to the 10 kV line to enhance the reliability of the power supply. Here, two microgrids and one battery are presented as an example. Microgrid 1 is grid-connected during normal operation, whereas Microgrid 2 operates in an islanded mode during normal operation. Their specific structures are, respectively, shown in Figure 2 and Figure 3.

Microgrid 1 is suitable for systems where the voltage of the external grid is relatively stable. During normal operation, circuit breaker B2 is closed, and the microgrid is connected to the 10 kV line. Microgrid 2 is suitable for remote mountainous areas, villages, islands, and other areas where power supply conditions are very severe and the external grid is unstable. During normal operation, circuit breaker B3 is disconnected, allowing this microgrid to operate independently. When it needs to be connected to the main grid, B3 can be closed for grid-connected operation.

## 3. Design of Grid-Connected/Islanded AC/DC Hybrid Microgrid Structures

### 3.1. Topology and Control Strategy of the Grid-Connected Microgrid (Microgrid 1)

The topology of the grid-connected microgrid designed in this paper is depicted in Figure 2. The microgrid incorporates AC and DC buses at three distinct voltage levels: a 380 V AC bus, a 300 V low-voltage DC bus, and an 800 V high-voltage DC bus. The reasons for choosing 300 V as the operating voltage of the low-voltage DC bus are as follows: Taking into comprehensive consideration the factors of insulation withstand voltage, cost, and power supply reliability, the operating voltage *U_DC_* of the low-voltage DC bus has been set at 300 V, which is lower than the peak value of the 220 V single-phase AC source voltage.

The 380 V AC bus derives its voltage from the 10 kV distribution line via the transformer T3. The PV module consists of multiple PV units interconnected in series and parallel, and the function of grid connection is realized through a two-stage circuit. The first stage is a Boost converter, and it is mainly used for the Maximum Power Point Tracking (MPPT) of the PV module. The control algorithm adopted is the incremental conductance method [36]. The second stage is a two-level inverter, and it implements constant DC voltage control. Its topology and control structure are shown in Figure 4, with its mathematical model presented in Equation (1).
(1)$$\left\{\begin{array}{l} i_{dref}=\left(U_{dref}-U_{dc}\right)\cdot \left(K_{pu}+\frac{K_{iu}}{s}\right)\\ i_{qref}=0\\ u_{d}=\left(i_{dref}-i_{d}\right)\cdot \left(K_{pi}+\frac{K_{ii}}{s}\right)+e_{d}-\omega L\cdot i_{q}\\ u_{q}=\left(i_{qref}-i_{q}\right)\cdot \left(K_{pi}+\frac{K_{ii}}{s}\right)+e_{q}+\omega L\cdot i_{d} \end{array}\right.$$

The grid-connected system of the BT module is similar to that of the PV module, also interfacing with the grid through a two-stage circuit. However, there are two differences. Firstly, the bidirectional Boost converter adopts constant voltage control. Its topology and control structure are shown in Figure 5, with its mathematical model presented in Equation (2). Secondly, the two-level inverter adopts constant power control, and the reference and sampling values of the d-axis represent active power values. The AC loads can be energized or de-energized according to the actual situation.
(2)$$\left\{\begin{array}{l} i_{ref}=\left(U_{dcref}-U_{dc}\right)\cdot \left(K_{pu}+\frac{K_{iu}}{s}\right)\\ u_{d}=\left(i_{refl}-i_{dc}\right)\cdot \left(K_{pi}+\frac{K_{ii}}{s}\right) \end{array}\right.$$

The 300 V low-voltage DC bus is connected to the AC bus via a transformer and a two-level converter. The two-level converter adopts constant current control and enables power transfer in both directions. Both the PV and BT modules are connected to the DC bus through a DC/DC converter. The converter of the BT module adopts constant voltage control as shown in Figure 5 to maintain the stability of the DC bus voltage, while the converter of the PV module realizes the MPPT function the same as the AC side. The WT module is connected to the DC bus through a two-level rectifier, and it adopts constant power control. The loads on the low-voltage DC bus are categorized into conventional loads and critical loads. The conventional loads can be energized or de-energized depending on the actual situation, while the critical loads in hospitals and government departments require a stable power supply. The topology of the critical load system is illustrated in Figure 6a. The terminals a, b, c, and n constitute a three-phase, four-wire 380 V/220 V AC load system with a rated frequency of 50 Hz. The block diagram for the SPWM signal of the critical load system is shown in Figure 6b. The mathematical model corresponding to Figure 6b is presented in Equation (3).
(3)$$i_{ref}=\left(U_{ref}-U_{rms}\right)\cdot \frac{1}{U_{ref}}\cdot \left(K_{pu}+\frac{K_{iu}}{s}\right)$$

With the advancement of electric vehicle charging technology, 800 V high-voltage fast charging technology has emerged [37], and it helps to improve the charging efficiency of electric vehicles. To facilitate fast charging, an 800 V high-voltage DC bus is incorporated into this microgrid. The high-voltage and low-voltage DC buses are interconnected through a Boost converter, which adopts constant voltage control.

### 3.2. Topology and Control Strategy of the Islanded Microgrid (Microgrid 2)

The topology of the islanded microgrid designed in this paper is shown in Figure 3, and the voltage levels of the AC and DC buses are consistent with those in the grid-connected microgrid.

The BT, PV, and WT modules connected to the 300 V low-voltage DC bus are controlled in the same way as those in the grid-connected microgrid. The 800 V high-voltage DC bus is connected to the low-voltage DC bus through a Boost converter, and electric vehicle charging piles can be installed on the bus to realize fast charging. The converter between the AC bus and the DC bus adopts droop control. The topology and control structure of droop control are illustrated in Figure 7. The expressions of active power and reactive power are presented in Equation (4), and the expressions of active and reactive power droop coefficients are presented in Equation (5). The subscript n in Equation (5) indicates the rated value. When the AC loads are energized or de-energized, the converter will quickly adjust the output power to meet the requirements of the loads.
(4)$$\left\{\begin{array}{l} P=\frac{3}{2}\cdot \left(V_{d}\cdot I_{d}+V_{q}\cdot I_{q}\right)\\ Q=\frac{3}{2}\cdot \left(V_{q}\cdot I_{d}-V_{d}\cdot I_{q}\right) \end{array}\right.$$
(5)$$\left\{\begin{array}{l} m=\frac{w_{n}}{P_{n}}\times 1\%\\ n=\frac{U_{n}}{Q_{n}}\times 5\% \end{array}\right.$$

## 4. Control Strategy for Fast Access to the Backup Power

The novel microgrid cluster structure shown in Figure 1 is capable of rapidly restoring power supply in the event of a power source failure. The sequence of actions taken by the components within this structure following a failure is described in detail in this section. The main power source failure scenarios discussed in this paper include single line-to-ground short-circuit faults, line-to-line short-circuit faults, double line-to-ground short-circuit faults, three-phase short-circuit faults, and other short-circuit faults, as well as single line disconnection faults, double line disconnection faults, and other disconnection faults. The occurrence of these faults will trigger the relay protection switches, rendering the affected power source unable to continue supplying power, thereby causing great harm to the safety and stability of the power system.

When the main lines are connected to the 10 kV section I (or Ⅱ) bus and the power source on the grid side fails, the circuit breaker B1 will be disconnected. At this point, the thyristor switch composed of two anti-parallel thyristors (which were in the hot standby state) will be quickly triggered, and the main lines will be supplied by the 10 kV section Ⅱ (or I) bus to ensure the power supply to the critical loads. Subsequently, the disconnecting switch K1 (or K2) is disconnected, K2 (or K1) is closed, and then the circuit breaker B1 is reclosed. The thyristor switch will stop being triggered, and the disconnecting switch connected to it will switch to the 10 kV section I (or Ⅱ) bus. It will be in a hot standby state when the power source of the 10 kV section I (or Ⅱ) bus is restored. The specific control strategy is illustrated in Figure 8.

The trigger signal for the thyristor switch is shown in Figure 9. Signal 1 represents the trigger signal of the circuit breaker B1, the value of which is 1 when B1 is triggered on and 0 when it is triggered off. Signal 2 denotes the outlet current signal of circuit breaker B1. When the current value falls below the set threshold, the signal value is 0, and vice versa, it is 1.

## 5. Evaluation Indicators

Stability: In order to ensure that the system can provide highly reliable power to loads, the switching process after a fault occurs must be rapid. Additionally, the voltage fluctuation within each microgrid and the microgrid cluster must adhere to National Standards of the People’s Republic of China GB/T 12325—2008 Power Quality—Deviation of Supply Voltage Standard. Specifically [38], the maximum permissible deviation for the three-phase power source voltage of 10 kV and below should not exceed 7% of the nominal system voltage. Furthermore, the voltage fluctuation range of the DC bus is typically required to be within 5% of the nominal voltage.

Power quality: The existence of harmonics can cause great harm to the power system, so it is also imperative to comply with National Standards of the People’s Republic of China GB/T 14549—1993 Quality of electric energy supply Harmonics in public supply network standard [39]. This standard stipulates that the total harmonic distortion (THD) should be less than 5% for 380 V systems and less than 4% for 10 kV systems. Moreover, the THD of current is generally required to be below 5%.

## 6. Case Study

In this section, we present a series of case simulation results to validate the feasibility of the proposed new microgrid cluster, grid-connected microgrid, and islanded microgrid structures and to compare them with existing studies.

### 6.1. Case Study of the Grid-Connected Microgrid Structure

The parameters of the grid-connected microgrid structure are shown in Table 1.

In the grid-connected microgrid, the sun irradiance of the PV module on the AC side changes from 1050 W/m^2^ to 800 W/m^2^ from 1 s to 1.5 s, while that of the PV module on the DC side is set at 800 W/m^2^. The simulation step is chosen to be 1 × 10^−6^, and the specific simulation results are shown below.

#### 6.1.1. Simulation Results on the AC Side

The waveforms of the output voltage and current of the PV module on the AC side are shown in Figure 10. In this figure, the voltage waveform remains stable, and the current decreases with the decrease in sun irradiance, with the amplitude changing from 32.12 A to 24.86 A.

The waveforms of SOC, current, and terminal voltage of the AC-side BT module are shown in Figure 11. Before 1.234 s, the battery is in the charging state, the SOC increases, and the output current is negative; after 1.234 s, the battery is in the discharging state, the SOC decreases, the output current is positive, and the terminal voltage also decreases.

The power, grid-connected voltage, and current waveforms generated by the joint operation of PV and BT modules are shown in Figure 12a,b. The active power value is 13.5 kW, and the reactive power value is 0 Var, which is consistent with the set value. Additionally, the waveforms of the voltage and current are stable without any obvious distortion. 

The voltage deviation and THD of both voltage and current for the joint operation of PV and BT modules on the AC side are, respectively, shown in Table 2 and Table 3. Comparison with the indicators in Section 5 reveals that the voltage and current comply with the requirements of the relevant standards.

#### 6.1.2. Simulation Results of the AC–DC Interface Line

The power, voltage, and current waveforms of the AC–DC interface line are shown in Figure 13. The power flow direction is from the AC side to the DC side. The active power value is about 2.3 kW, and the reactive power value is 0 Var. The waveforms of the voltage and current are stable without any obvious distortion.

The voltage deviation and THD of both voltage and current of the AC–DC interface line are shown in Table 4 and Table 5. The values comply with the requirements of the relevant standards.

#### 6.1.3. Simulation Results on the DC Side

The output active power waveform of the PV module on the DC side is shown in Figure 14, and the stable output active power value is 12 kW.

The waveforms of SOC, current, and terminal voltage of the BT module on the DC side are shown in Figure 15, and it can be seen that the battery is in the charging state.

The active power waveform of the WT module connected to the DC side is shown in Figure 16, and the stable output active power value is 25 kW.

The voltage waveform of the low-voltage DC bus is shown in Figure 17, indicating that the stabilized voltage value is equal to the set value of 300 V.

The phase and line voltage waveforms of the critical loads connected to the low-voltage DC bus are shown in Figure 18. The voltage waveforms are stable, all three-phase symmetrical, with no obvious fluctuations or distortions.

The voltage deviation and THD of the phase voltages of the critical loads are presented in Table 6, and the values comply with the requirements of the relevant standards.

The voltage waveform of the high-voltage DC bus is shown in Figure 19, and the stable voltage value is 800 V.

### 6.2. Case Study of the Islanded Microgrid Structure

Compared with the grid-connected microgrid, the AC loads and the control strategy of the VSC converter between the AC and DC sides are different, and the parameters of the AC load are presented in Table 7.

As shown in Table 7, in the islanded microgrid, the active power value of the AC loads is 10 kW from 0 s to 1 s, increases to 14 kW at 1 s, and then changes back to 10 kW at 1.5 s, and the reactive power value is 1 kVar from 0 s to 1 s, increases to 1.5 kVar at 1 s, and then changes back to 1 kVar at 1.5 s. The simulation step is set at 1 × 10^−6^ s, and the specific simulation results are shown below.

#### 6.2.1. Simulation Results on the DC Side

The active power waveform of the PV module on the DC side is shown in Figure 20, and the stable power value is 12 kW.

The active power waveform of the WT module on the DC side is shown in Figure 21, and the stable power value is 25 kW.

The waveforms of SOC, current, and terminal voltage of the BT module on the DC side are shown in Figure 22, which shows that the battery is in the charging state.

The voltage waveform of the low-voltage DC bus on the DC side is shown in Figure 23, and the voltage value, when stabilized, is equal to the set value of 300 V.

The phase and line voltage waveforms of the critical loads connected to the low-voltage DC bus are shown in Figure 24. The voltage waveforms are stable, all three-phase symmetrical, with no obvious fluctuations or distortions.

The voltage deviation and THD of the phase voltages of the critical loads are given in Table 8, and the values comply with the requirements of the relevant standards.

The voltage waveform of the high-voltage DC bus is shown in Figure 25, with a stable voltage value of 800 V.

#### 6.2.2. Simulation Results on the AC Side

The power waveform output from the converter on the AC side is shown in Figure 26. The active power value is 10 kW from 0 s to 1 s, then increases to 13.5 kW at 1 s, and subsequently returns to 10 kW after 1.5 s. Similarly, the reactive power value is 1 kVar from 0 s to 1 s, increases to 1.44 kVar at 1 s, and then returns to 1 kVar after 1.5 s. It is evident that with droop control, the power output from the converter changes after a sudden change in the loads, but there is still some error.

The frequency waveform on the AC side is shown in Figure 27. The frequency decreases to 49.83 Hz from 1 s to 1.5 s as the active power of the load increases, and it is consistent with the droop characteristic of active power frequency.

The voltage and current waveforms on the AC side are shown in Figure 28. After a sudden change in the AC loads, the voltage magnitude decreases from 311 V to 304.3 V at 1 s, which is in accordance with the droop characteristic of reactive power voltage, whereas the current magnitude increases from 21.65 A to 29.67 A.

The voltage deviation and THD of voltage and current on the AC side are shown in Table 9 and Table 10, respectively. The values comply with the requirements of the relevant standards.

### 6.3. Case Study for the Structure for Fast Access to the Backup Power

For the novel microgrid cluster structure shown in Figure 1, the models and parameters of grid-connected and islanded microgrids have been introduced in the previous section, and the rest of the parameters are shown in Table 11.

In the novel microgrid cluster structure, following the control strategy in Figure 7, the following case is presented: The voltage of the power source is 35 kV. From 0 s to 1 s, the system is supplied by the 10 kV section I bus. At 1 s, the power source connected to the section I bus fails, causing the circuit breaker B1 to disconnect and the thyristor switch to trigger. From 1 s to 2 s, the system is supplied by the power source connected to the 10 kV section Ⅱ bus via the thyristor switch. At 2 s, the disconnecting switch K2 is closed, K1 is disconnected, the circuit breaker B1 is closed, and the thyristor switch stops being triggered. The disconnected switches on both sides of the thyristor switch are switched to the 10 kV section I bus. The simulation step is set at 1 × 10^−6^ s, and the simulation results are as follows:

The trigger signal waveforms of the thyristor switch are shown in Figure 29. It can be seen that there is a difference in the trigger time of the three-phase thyristors, but they all turn off at the same time when switching off.

The voltage and current waveforms at the outlet of circuit breaker B1 are shown in Figure 30. The voltage experiences a short transition process from 1 s to 1.02 s, after which it returns to normal levels. The current is interrupted from 1 s to 2 s and subsequently returns to normal levels after 2 s.

The voltage deviation and THD of voltage and current at the outlet of circuit breaker B1 are shown in Table 12 and Table 13, respectively. The values comply with the requirements of the relevant standards.

The voltage and current waveforms at the outlet of the thyristor switch are shown in Figure 31. The voltage experiences a short interruption from 1 s to 1.02 s, subsequently returning to normal levels. The current begins to increase at 1.02 s and decreases to 0 after 2 s.

The voltage deviation and THD of voltage and current at the outlet of the thyristor switch are shown in Table 14 and Table 15, respectively. The values comply with the requirements of the relevant standards.

The voltage and current waveforms at the outlet of the 10 kV bus are shown in Figure 32. These waveforms are only briefly interrupted from 1 s to 1.02 s.

The voltage deviation and THD of voltage and current at the outlet of the 10 kV bus are shown in Table 16 and Table 17, respectively. The values comply with the requirements of the relevant standards.

### 6.4. Case Study for Comparisons with the Existing Research

#### 6.4.1. Innovative Structure for Fast Access to the Backup Power

In this paper, taking advantage of the characteristics of thyristor switches that can respond quickly, an innovative structure that allows backup power to be accessed quickly during faults is designed. The structure is shown in Figure 1. And in Section 6.3, a specific simulation case is carried out for the study.

Among the existing studies, if the thyristor switch is not used, a circuit breaker is required to be used directly to access the backup power after a fault occurs. According to the International Electrotechnical Commission (IEC) and national standards, the closing time of high-voltage circuit breakers is no more than 0.2–0.3 s. Therefore, the simulation case is designed as follows: The complete closing time of circuit breaker B, which replaces the thyristor switch, is set at 1.2 s, and other parameters are the same as those in Section 6.3. The main simulation results are shown in Figure 33, Figure 34 and Figure 35.

Comparing Figure 30, Figure 31 and Figure 32 with Figure 33, Figure 34 and Figure 35, it can be seen that, according to the existing studies, when using circuit breakers instead of thyristor switches to access the backup power, the duration of the power outage is too long. This prolonged power outage will cause harm to the equipment and seriously affect the stability of the power system. It is undesirable for critical loads that have high requirements for power supply reliability. Therefore, the innovative structure proposed in this paper is better than the existing studies and has high practical value.

#### 6.4.2. Innovative Structure of Three-Phase Four-Wire Critical Load System

The three-phase, four-wire structure, as shown in Figure 6a, is used for the critical loads in this paper. It is able to adapt to the changes in the operating conditions of the loads. In this subsection, taking the grid-connected microgrid as an example, the following case simulation is designed: Loads of the three phases are set as follows: *R_a_* = 30 Ω, *R_b_* = 40 Ω, and *R_c_* = 35 Ω. Additionally, the load of phase B is cut off at 1.5 s. Other parameters are the same as those in Section 6.1. The main simulation results are as follows:

As we can see from Figure 36, the phase voltages and line voltages of the loads can remain stable despite the three-phase load imbalance before 1.5 s. After 1.5 s, the load of phase B is cut off, and its voltage becomes 0, while the voltages of phase A and phase C remain stable, and the line voltage *U_AC_* is still stable. Compared with the existing studies, when only three single-phase inverters are used, the line voltages cannot be provided; when three-phase inverters are used, the cost increases, and they are only applicable to the condition of symmetrical three-phase loads. In addition, as shown in Figure 37, the DC bus voltage remains stable without fluctuation during the process, which demonstrates the stability of the structure.

#### 6.4.3. Innovative Structure of the 800 V High-Voltage DC Bus

In this paper, in order to meet the need for the rapidly developing electric vehicle charging technology, an 800 V high-voltage DC line is included in both grid-connected and islanded microgrid structures. This structure can effectively reduce the harmonic content of the grid and improve system reliability.

For the existing structure, as shown in Figure 38, the electric vehicle charging piles are usually connected to the AC grid through converters, and the existence of the converters will make the harmonic content of the grid increase. In this paper, a simulation case of different numbers of converters connected to a 380 V AC microgrid is carried out, and the harmonic content detection of the grid current is performed. The THD of the AC microgrid current is shown in Table 18. From the data in Table 18, it can be seen that with the increase in the number of converters, the harmonic content in the grid is increasing rapidly, which shows that the 800 V DC bus designed in this paper is of great significance.

## 7. Cost and Efficiency Analysis of the Proposed Innovative Structures

The innovative structure proposed in this paper allows backup power to be accessed quickly during faults. This can greatly reduce the outage time, avoid serious harm to the equipment in the system, and guarantee normal industrial production. Therefore, it can help reduce losses and costs. Additionally, the control strategy is simple and can improve the efficiency of the operation.

For the grid-connected AC/DC hybrid microgrid structure, the PV and BT modules are retained on the AC side of the structure considering the presence of existing rooftop PV and distributed BT modules, and they have already been integrated into AC microgrids. Our structure is designed to be cost-effective, so the new structure can be constructed based on the existing AC microgrids, hence retaining this configuration. At the same time, the newly built distributed energy, such as PV, BT, and WT modules, should ideally be located on the low-voltage DC bus. This design results in fewer converters being used in the system. On the one hand, energy losses can be reduced, and efficiency can be improved; on the other hand, construction costs can be reduced. In addition, the presence of the 800 V high-voltage DC bus can also reduce the number of inverters, which is effective in improving efficiency and reducing costs.

For the islanded AC/DC hybrid microgrid structure, since it is mainly used in remote areas with few existing distributed PV and BT modules, all kinds of distributed energy are connected to the low-voltage DC bus in order to reduce costs and improve efficiency.

## 8. Conclusions

In summary, the novel microgrid cluster structure proposed in this paper can realize fast switching to backup power after the loss of a power source. The structure takes advantage of the characteristics of the thyristor switch, which can respond quickly with a short transition process and fast switching speed after a fault occurs. Moreover, it is equipped with microgrids and batteries, which can effectively improve the reliability of the power supply for critical loads. In addition, two types of AC/DC hybrid microgrid structures in grid-connected and islanded states are designed in this paper according to the actual conditions, both of which contain three AC/DC voltage levels. The DC voltage is divided into two levels according to the needs of the fast charging of electric vehicles and the power supply for critical loads. The above two AC/DC hybrid microgrid structures can not only achieve a high proportion of distributed energy consumption but also have simple control, high stability, and high power quality, which can reduce costs and improve efficiency.

There are still some limitations in the application of the study performed in this paper that need to be further explored in depth: 1. The thyristor switch used in this paper needs to withstand a voltage value of more than 10 kV, which needs to be realized by using high-voltage thyristor series technology, and the thyristor voltage equalization issue, as well as the synchronicity issue of the thyristor conduction and shutdown, should be considered. 2. In this paper, typical values of distributed power generation are used for simulation verification, while the effect of its volatility on the system should be considered in practical application. 3. When the load changes, there is an error between the power generated by the inverter and the actual load power in the islanded microgrid, so the control can be optimized to achieve better power tracking.

## Figures and Tables

**Figure 1 sensors-24-04778-f001:**
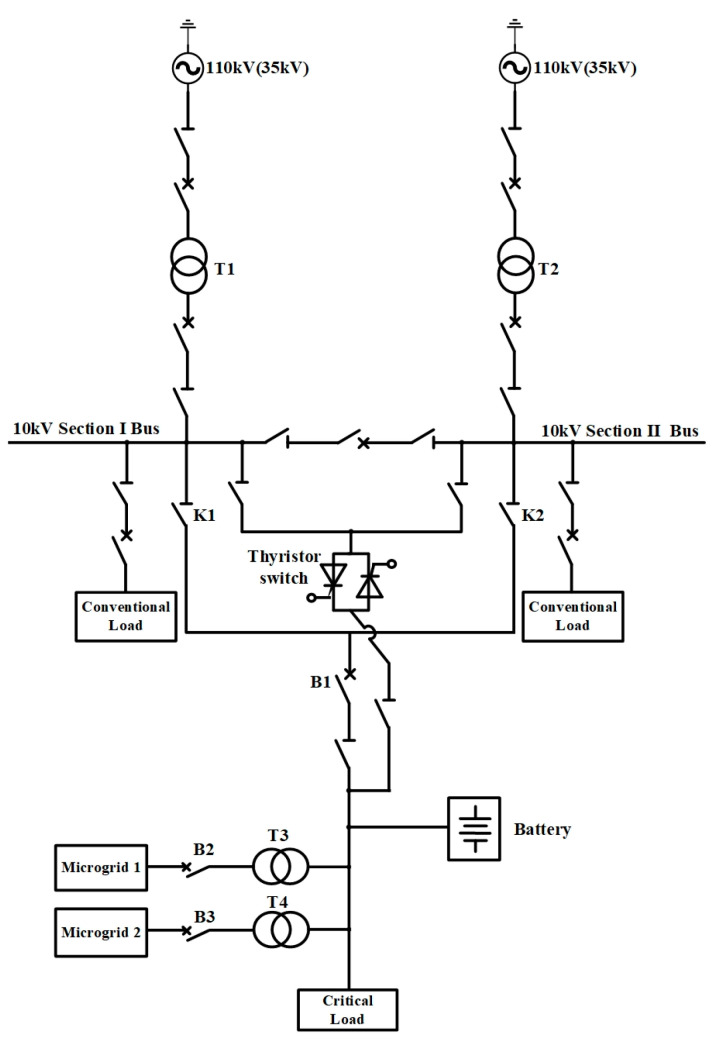
Topology of the novel AC/DC hybrid microgrid cluster.

**Figure 2 sensors-24-04778-f002:**
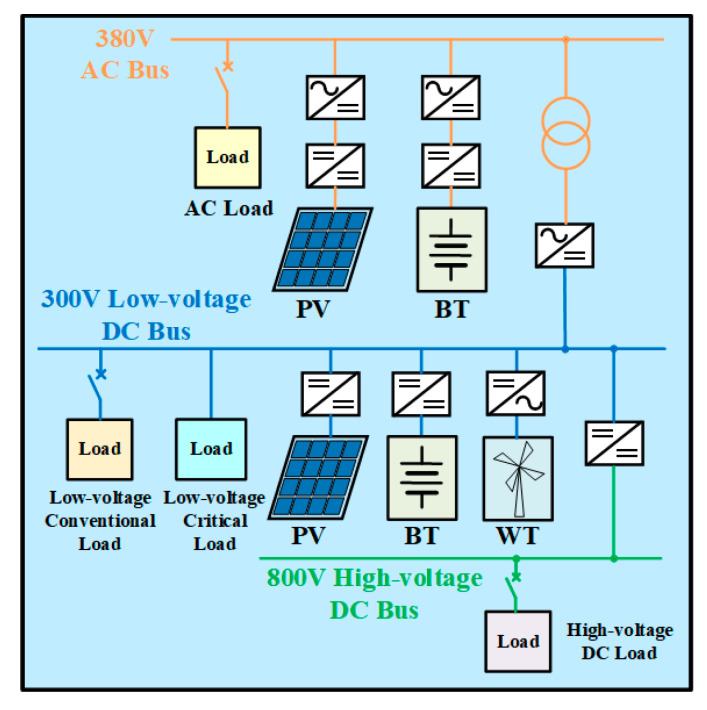
Topology of the grid-connected microgrid.

**Figure 3 sensors-24-04778-f003:**
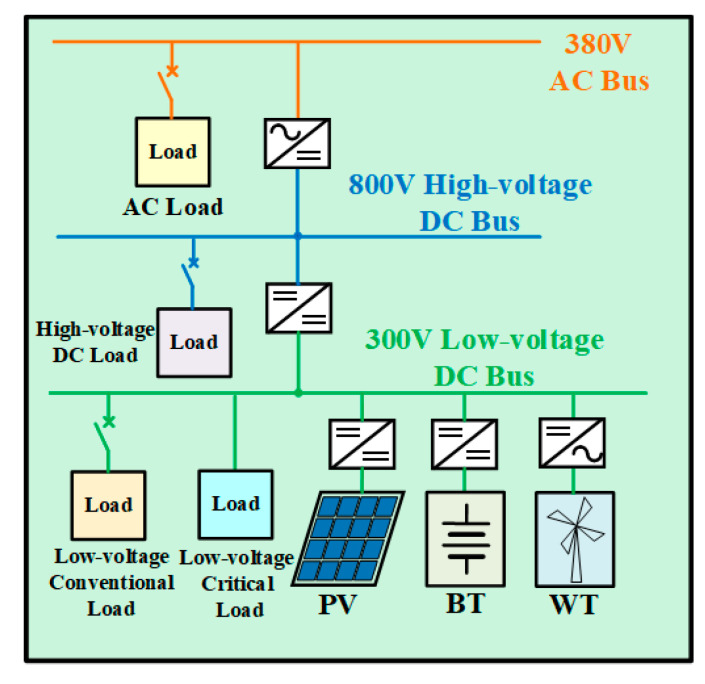
Topology of the islanded microgrid.

**Figure 4 sensors-24-04778-f004:**
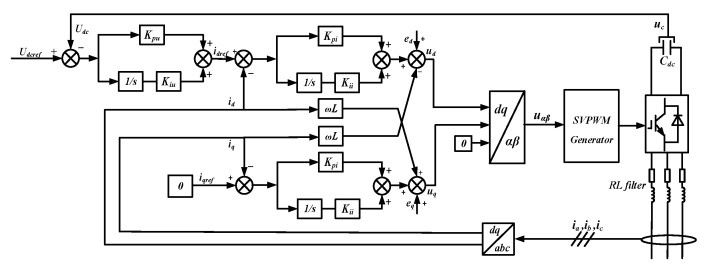
Block diagram of the two-level converter controller.

**Figure 5 sensors-24-04778-f005:**
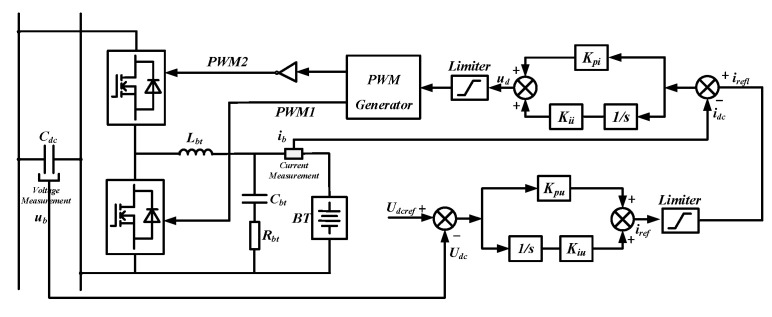
Block diagram of the DC/DC bidirectional converter controller.

**Figure 6 sensors-24-04778-f006:**
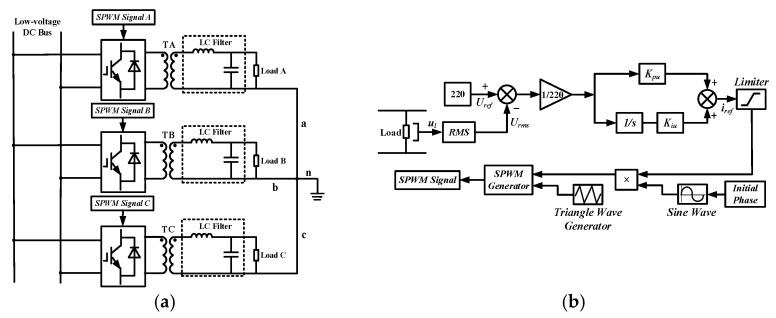
(**a**) Topology of the critical load system; (**b**) Block diagram for the SPWM signal of the critical load system.

**Figure 7 sensors-24-04778-f007:**
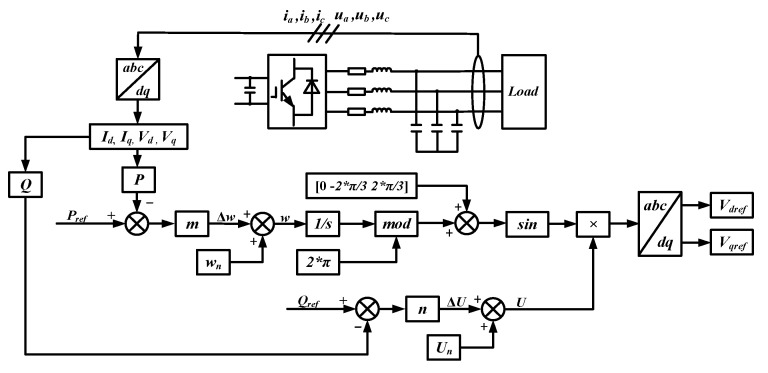
Block diagram of droop control.

**Figure 8 sensors-24-04778-f008:**
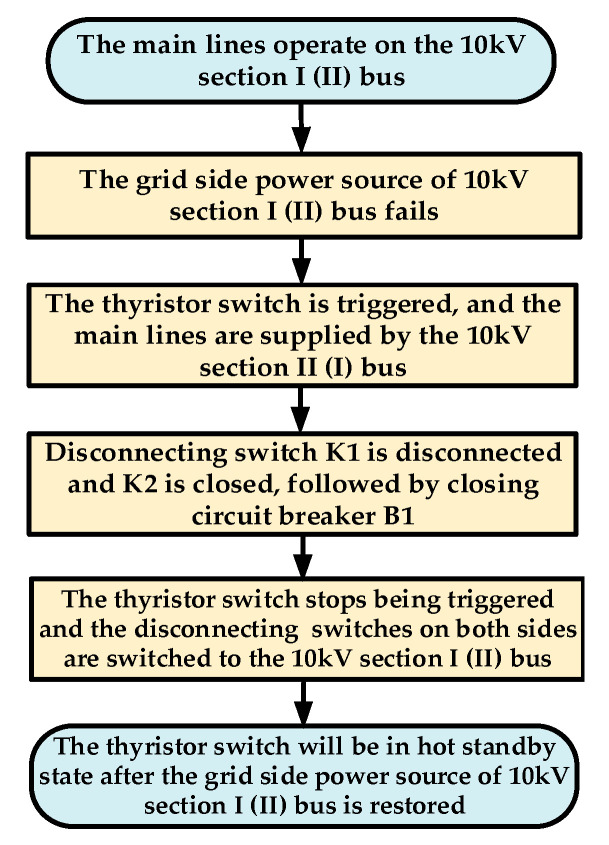
Control strategy for the novel microgrid cluster structure after faults.

**Figure 9 sensors-24-04778-f009:**
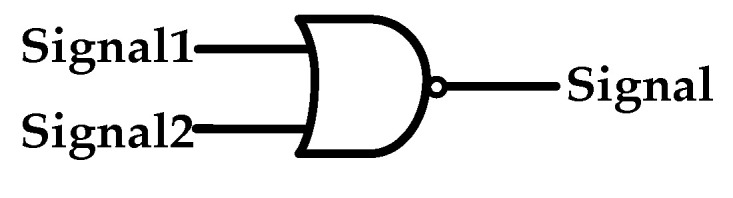
Trigger signal of the thyristor switch.

**Figure 10 sensors-24-04778-f010:**
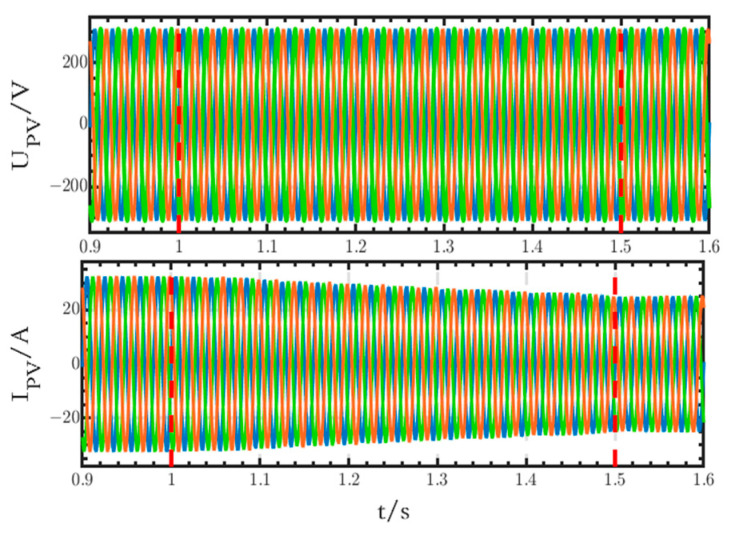
Voltage and current waveforms of the PV module on the AC side.

**Figure 11 sensors-24-04778-f011:**
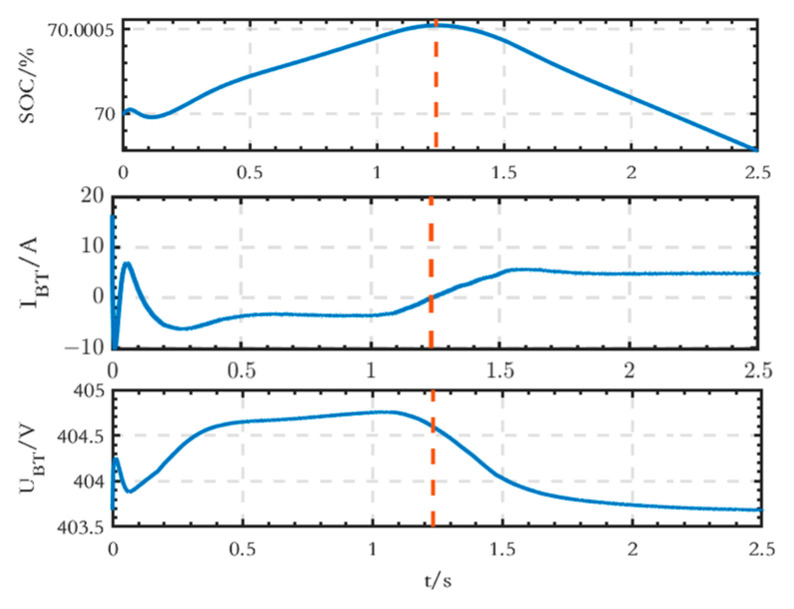
SOC, current, and terminal voltage waveforms of the BT module on the AC side.

**Figure 12 sensors-24-04778-f012:**
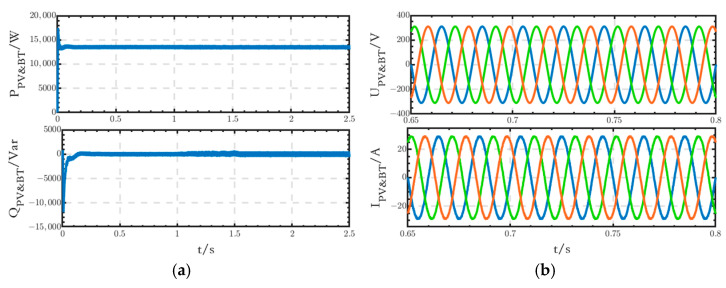
(**a**) Power waveforms generated by the joint operation of PV and BT modules; (**b**) voltage and current waveforms generated by the joint operation of PV and BT modules (In this paper, for phase voltages, the blue, green and red waveforms represent phase A, phase B and phase C respectively; for line voltages, the blue, green and red waveforms represent the line voltage between phase A and phase B, phase B and phase C, and phase C and phase A respectively).

**Figure 13 sensors-24-04778-f013:**
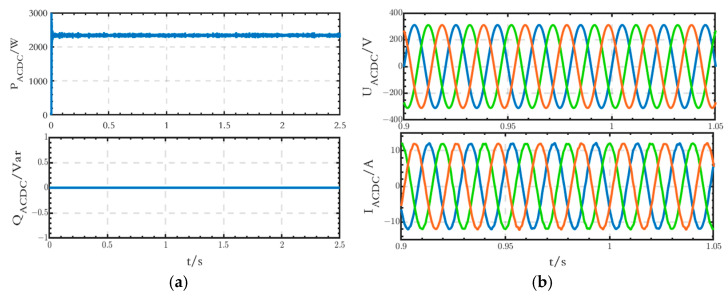
(**a**) Power waveforms of the AC–DC interface line; (**b**) voltage and current waveforms of the AC–DC interface line.

**Figure 14 sensors-24-04778-f014:**
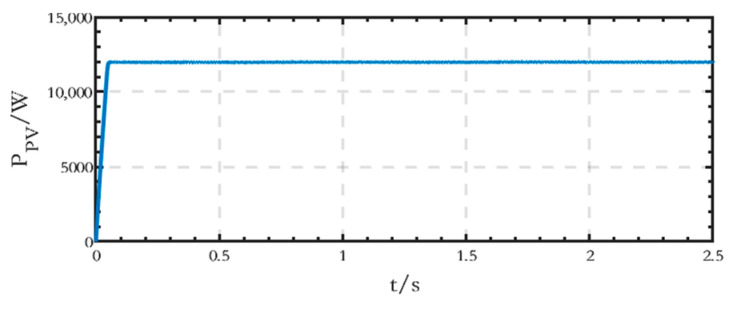
Active power waveform generated by the PV module on the DC side of the grid-connected microgrid.

**Figure 15 sensors-24-04778-f015:**
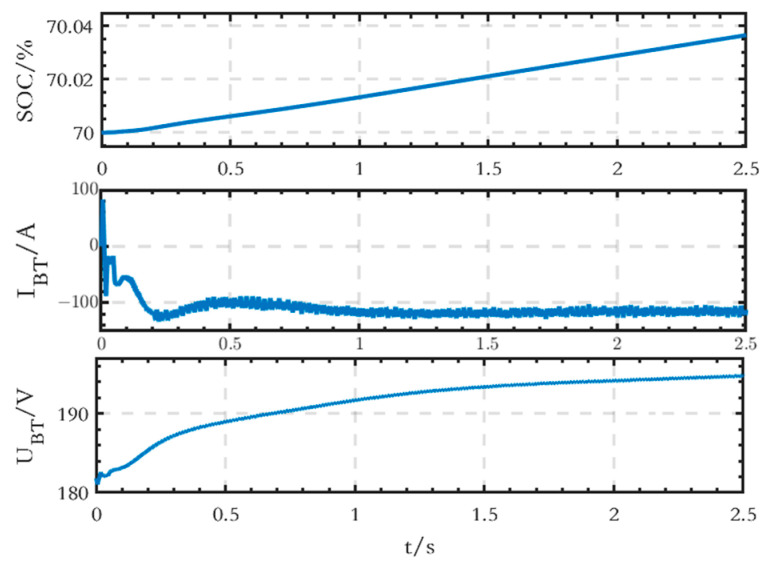
SOC, current, and terminal voltage waveforms of the BT module on the DC side of the grid-connected microgrid.

**Figure 16 sensors-24-04778-f016:**
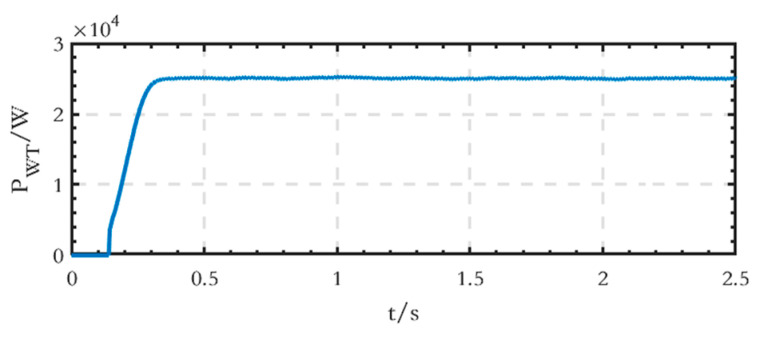
Active power waveform generated by the WT module on the DC side of the grid-connected microgrid.

**Figure 17 sensors-24-04778-f017:**
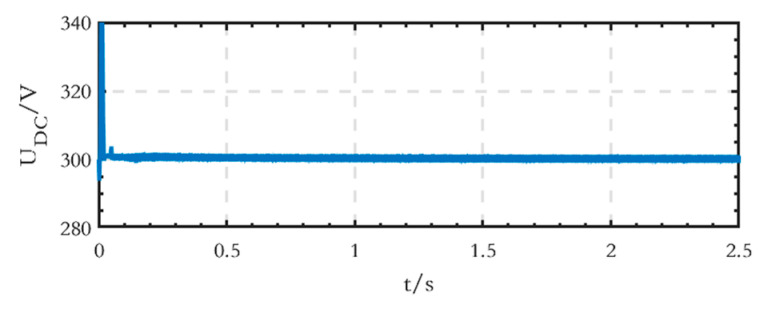
Voltage waveform of the low-voltage DC bus of the grid-connected microgrid.

**Figure 18 sensors-24-04778-f018:**
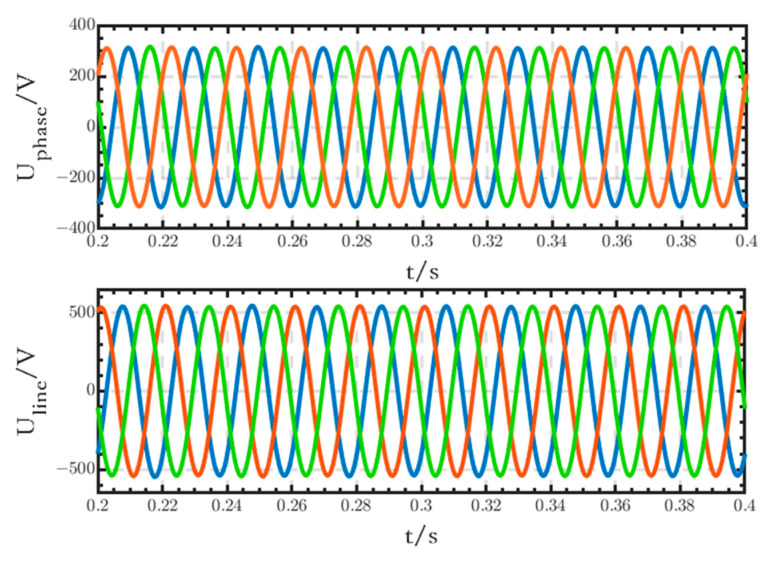
Phase and line voltage waveforms of the critical loads.

**Figure 19 sensors-24-04778-f019:**
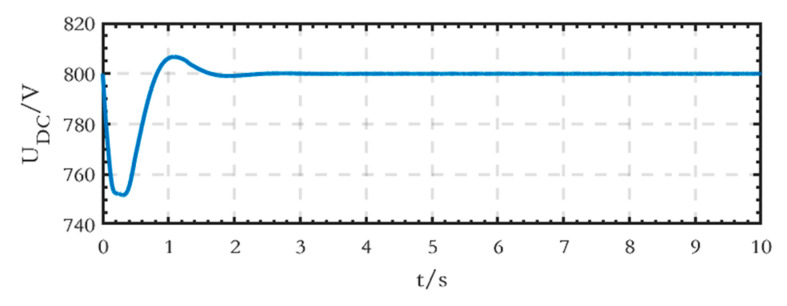
Voltage waveform of the high-voltage DC bus of the grid-connected microgrid.

**Figure 20 sensors-24-04778-f020:**
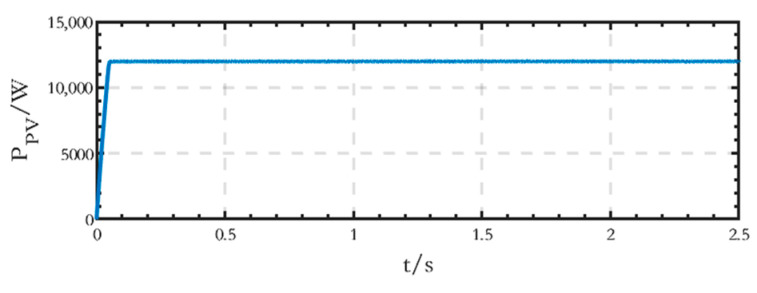
Active power waveform generated by the PV module on the DC side of the islanded microgrid.

**Figure 21 sensors-24-04778-f021:**
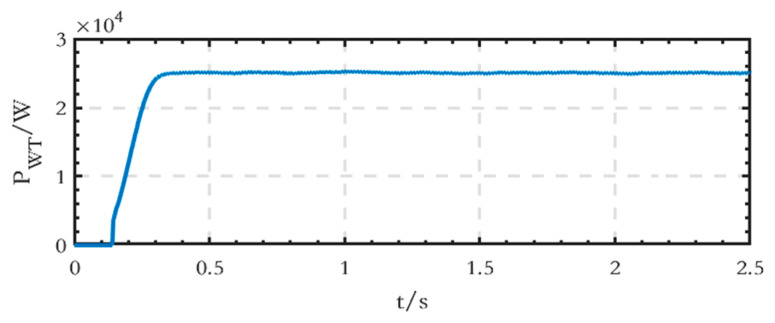
Active power waveform generated by the WT module on the DC side of the islanded microgrid.

**Figure 22 sensors-24-04778-f022:**
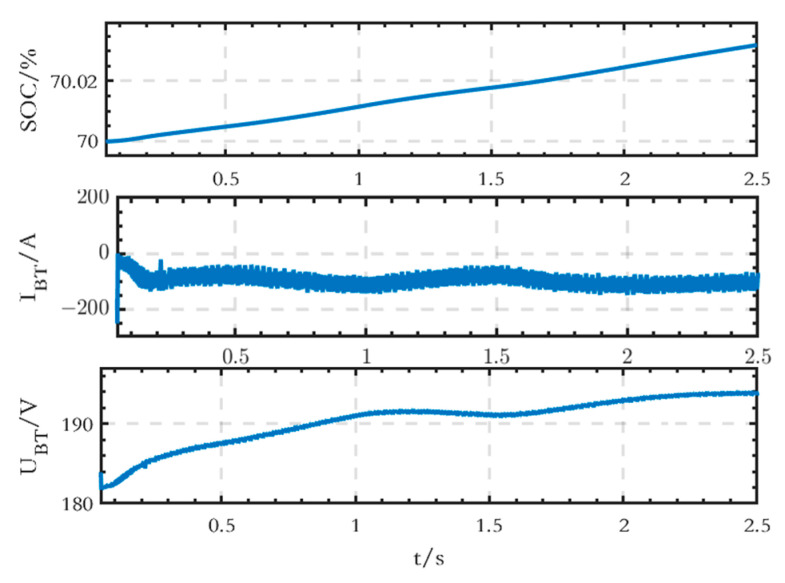
SOC, current, and terminal voltage waveforms of the BT module on the DC side of the islanded microgrid.

**Figure 23 sensors-24-04778-f023:**
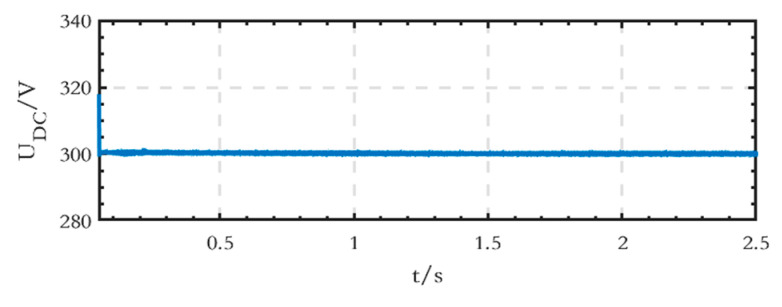
Voltage waveform of the low-voltage DC bus of the islanded microgrid.

**Figure 24 sensors-24-04778-f024:**
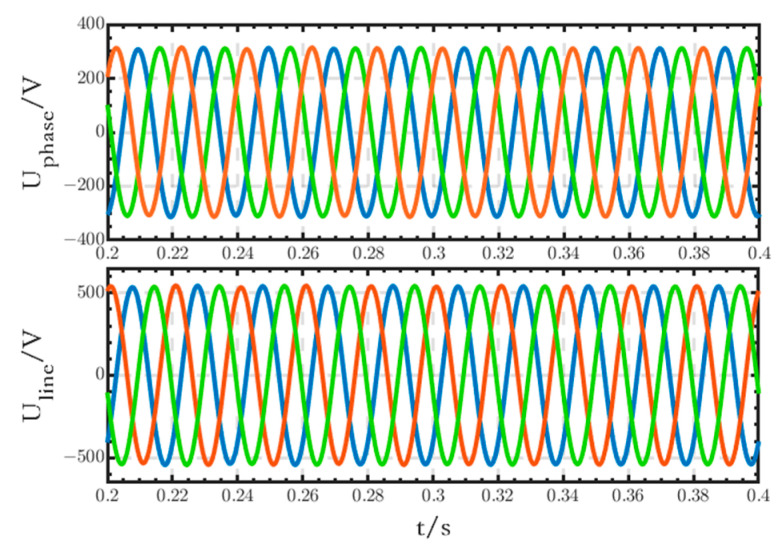
Waveforms of the phase and line voltages of critical loads.

**Figure 25 sensors-24-04778-f025:**
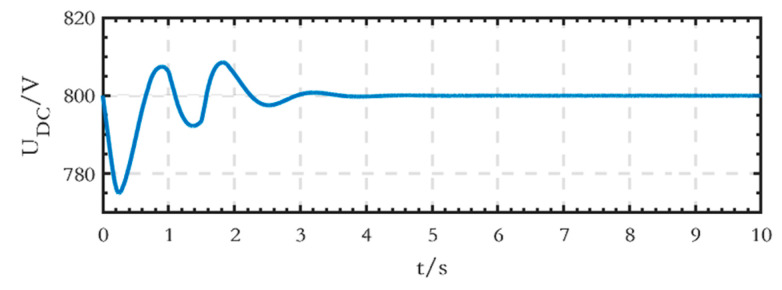
Voltage waveform of the high-voltage DC bus of the islanded microgrid.

**Figure 26 sensors-24-04778-f026:**
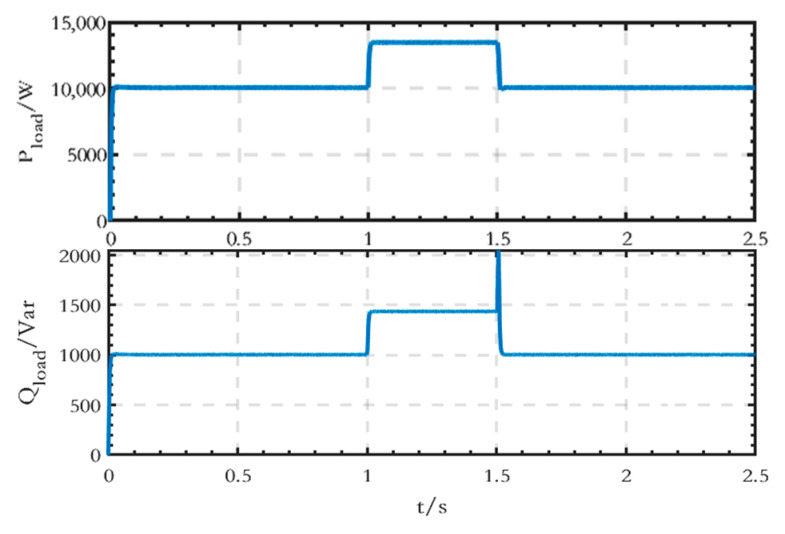
Power waveforms output from the converter.

**Figure 27 sensors-24-04778-f027:**
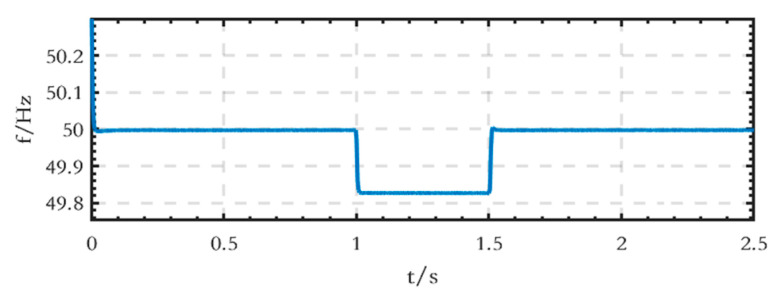
Frequency waveform on the AC side.

**Figure 28 sensors-24-04778-f028:**
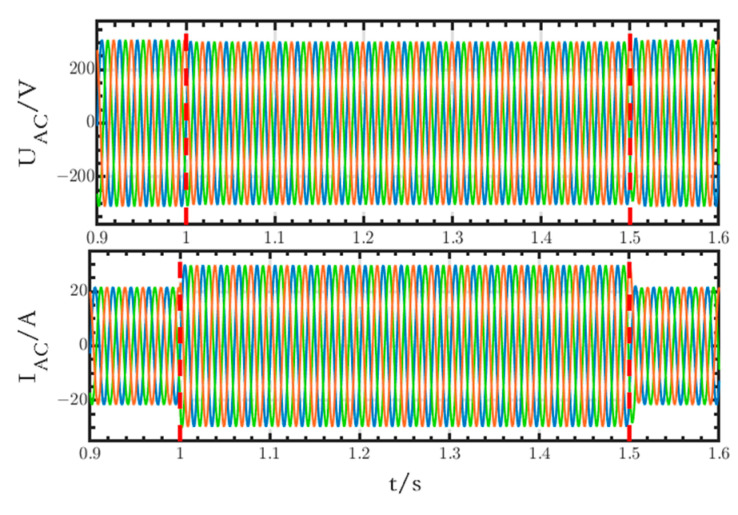
Voltage and current waveforms on the AC side.

**Figure 29 sensors-24-04778-f029:**
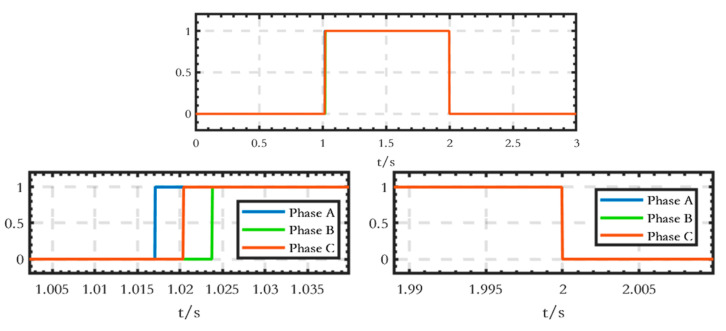
Trigger signal waveforms of the thyristor switch.

**Figure 30 sensors-24-04778-f030:**
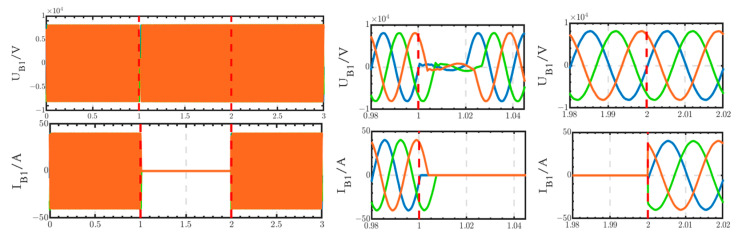
Voltage and current waveforms at the outlet of circuit breaker B1 when using the thyristor switch.

**Figure 31 sensors-24-04778-f031:**
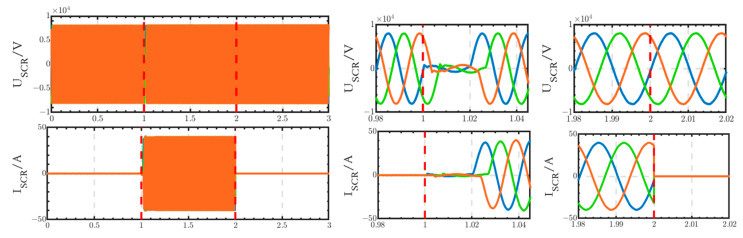
Voltage and current waveforms at the outlet of the thyristor switch.

**Figure 32 sensors-24-04778-f032:**
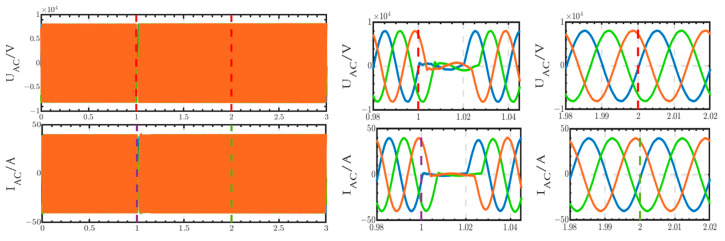
Voltage and current waveforms at the outlet of the 10 kV bus when using the thyristor switch.

**Figure 33 sensors-24-04778-f033:**
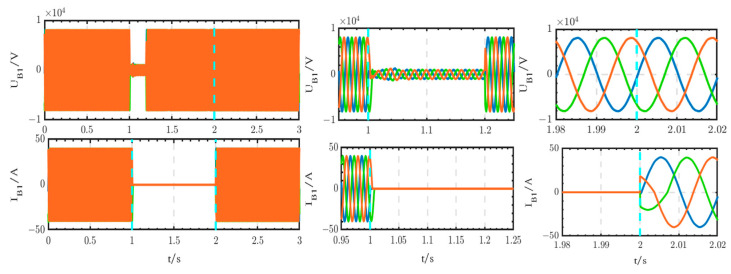
Voltage and current waveforms at the outlet of circuit breaker B1 when using circuit breaker B.

**Figure 34 sensors-24-04778-f034:**
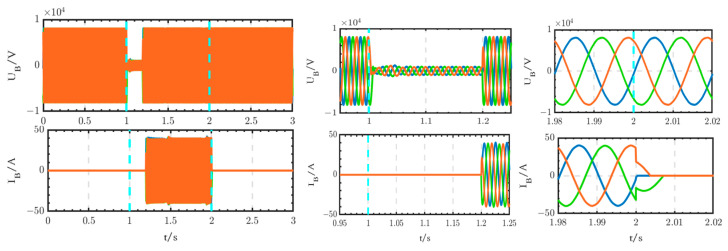
Voltage and current waveforms at the outlet of circuit breaker B.

**Figure 35 sensors-24-04778-f035:**
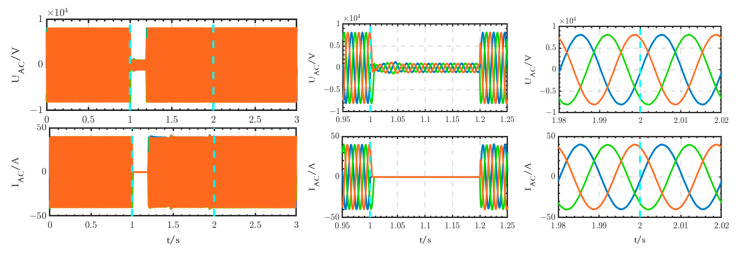
Voltage and current waveforms at the outlet of the 10 kV bus when using circuit breaker B.

**Figure 36 sensors-24-04778-f036:**
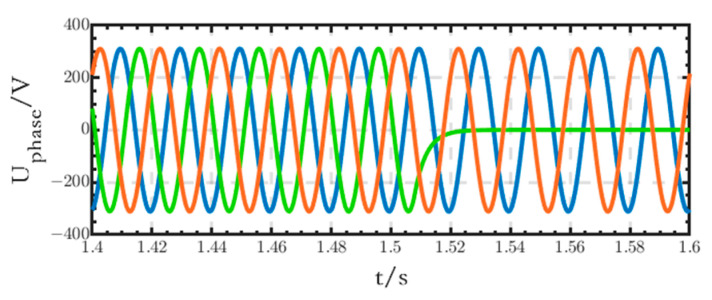
Phase and line voltage waveforms of critical loads.

**Figure 37 sensors-24-04778-f037:**
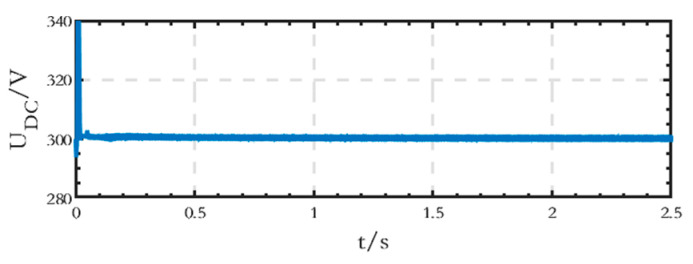
Voltage waveform of the low-voltage DC bus after cutting off the load of phase B.

**Figure 38 sensors-24-04778-f038:**
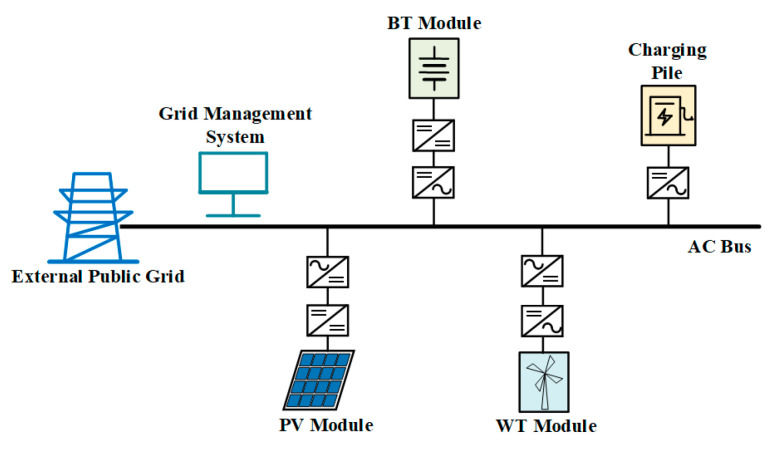
Structure of the existing AC microgrid.

**Table 1 sensors-24-04778-t001:** Parameters of the grid-connected microgrid structure.

Unit Name	Main Circuit Parameters	
Bus-AC	*U_AC_*	380 V
Bus-DC	*U_DC_h_*	800 V
	*U_DC_l_*	300 V
PV-AC Module	Sun Irradiance	1050/800 W/m^2^
	Cell Temperature	25 °C
BT-AC Module	*U_bt_ac_*	400 V
	Rated Capacity	200 Ah
	Response Time	1 s
	Initial SOC	70%
AC Load	*P_load_*	13.5 kW
	*Q_load_*	0 Var
Transformer between AC and DC circuit	Nominal Power	100 kVA
	Ratio	480/220 V
PV-DC Module	Sun Irradiance	800 W/m^2^
	Cell Temperature	25 °C
BT-DC Module	*U_bt_dc_*	180 V
	Rated Capacity	200 Ah
	Response Time	1 s
	Initial SOC	70%
WT Module	*v*	12 m/s
Critical Load	Transformer Ratio	300/305 V
	*R_l_* * _oad_ *	30 Ω

**Table 2 sensors-24-04778-t002:** Voltage deviation of *U_PV&BT_*.

	Voltage Deviation of *U_PV&BT_* (%)		
Time(s)	Phase A	Phase B	Phase C
0–1.0	0.27	0.27	0.27
1.0–1.5	0.27	0.27	0.27
1.5–2.5	0.27	0.27	0.27

**Table 3 sensors-24-04778-t003:** THD of *U_PV&BT_* and *I_PV&BT_*.

	THD of *U_PV&BT_* (%)			THD of *I_PV&BT_* (%)		
Time(s)	Phase A	Phase B	Phase C	Phase A	Phase B	Phase C
0–1.0	0.05	0.05	0.05	1.91	1.92	1.92
1.0–1.5	0.06	0.06	0.06	2.07	2.08	2.09
1.5–2.5	0.06	0.06	0.06	2.21	2.21	2.22

**Table 4 sensors-24-04778-t004:** Voltage deviation of *U_ACDC_* of the grid-connected microgrid.

	Voltage Deviation of *U_ACDC_* (%)		
Time(s)	Phase A	Phase B	Phase C
0–1.0	0.27	0.27	0.27
1.0–1.5	0.27	0.27	0.27
1.5–2.5	0.27	0.27	0.27

**Table 5 sensors-24-04778-t005:** THD of *U_ACDC_* and *I_ACDC_* of the grid-connected microgrid.

	THD of *U_ACDC_* (%)			THD of *I_ACDC_* (%)		
Time(s)	Phase A	Phase B	Phase C	Phase A	Phase B	Phase C
0–1.0	0.05	0.05	0.05	1.35	1.30	1.48
1.0–1.5	0.06	0.06	0.06	1.29	1.26	1.40
1.5–2.5	0.06	0.06	0.06	1.40	1.32	1.42

**Table 6 sensors-24-04778-t006:** Voltage deviation and THD of *U_phase_* of the grid-connected microgrid.

Voltage Deviation (%)			THD of *U_phase_* (%)		
Phase A	Phase B	Phase C	Phase A	Phase B	Phase C
0.06	0.10	0.10	0.07	0.07	0.07

**Table 7 sensors-24-04778-t007:** Parameters of the islanded microgrid structure.

Unit Name	Main Circuit Parameters		
AC Load	0–1.0 s	*P_load_*	10 kW
		*Q_load_*	1 kVar
	1.0–1.5 s	*P_load_*	14 kW
		*Q_load_*	1.5 kVar
	After 1.5 s	*P_load_*	10 kW
		*Q_load_*	1 kVar

**Table 8 sensors-24-04778-t008:** Voltage deviation and THD of *U_phase_* of the islanded microgrid.

Voltage Deviation (%)			THD of *U_phase_* (%)		
Phase A	Phase B	Phase C	Phase A	Phase B	Phase C
0.02	0.02	0.02	0.07	0.06	0.06

**Table 9 sensors-24-04778-t009:** Voltage deviation of *U_ACDC_* of the islanded microgrid.

	Voltage Deviation of *U_ACDC_* (%)		
Time(s)	Phase A	Phase B	Phase C
0–1.0	0.01	0.01	0.01
1.0–1.5	2.19	2.19	2.19
1.5–2.5	0.04	0.04	0.04

**Table 10 sensors-24-04778-t010:** THD of *U_ACDC_* and *I_ACDC_* of the islanded microgrid.

	THD of *U_ACDC_* (%)			THD of *I_ACDC_* (%)		
Time(s)	Phase A	Phase B	Phase C	Phase A	Phase B	Phase C
0–1.0	0.21	0.22	0.22	0.12	0.10	0.10
1.0–1.5	0.25	0.23	0.24	0.14	0.13	0.12
1.5–2.5	0.22	0.22	0.22	0.10	0.11	0.10

**Table 11 sensors-24-04778-t011:** Parameters of the novel microgrid cluster structure.

Unit Name	Main Circuit Parameters	
Bus-Source	*U_Source_*	35 kV
Transformer T1/T2	Nominal Power	1 MVA
	Ratio	35/10 kV
BT Module	*U_bt_dc_*	800 V
	Rated Capacity	200 Ah
	Response Time	1 s
	Initial SOC	70%
Conventional Load	*P_load_*	10 kW
	*Q_load_*	1 kVar
Critical Load	*P_load_*	500 kW
	*Q_load_*	50 kVar

**Table 12 sensors-24-04778-t012:** Voltage deviation of *U_B_*_1_.

	Voltage Deviation of *U_B_*_1_ (%)		
Time(s)	Phase A	Phase B	Phase C
0–1.0	1.32	1.32	1.32
1.0–2.0	0.98	0.99	0.98
2.0–3.0	0.77	0.77	0.77

**Table 13 sensors-24-04778-t013:** THD of *U_B_*_1_ and *I_B_*_1_.

	THD of *U_B_*_1_ (%)			THD of *I_B_*_1_ (%)		
Time(s)	Phase A	Phase B	Phase C	Phase A	Phase B	Phase C
0–1.0	0.38	0.38	0.38	0.36	0.36	0.36
1.0–2.0	0.43	0.44	0.43	--	--	--
2.0–3.0	0.44	0.46	0.45	0.43	0.44	0.45

**Table 14 sensors-24-04778-t014:** Voltage deviation of *U_SCR_*.

	Voltage Deviation of *U_SCR_* (%)		
Time(s)	Phase A	Phase B	Phase C
0–1	1.32	1.32	1.32
1–2	0.98	0.99	0.98
2–3	0.77	0.77	0.77

**Table 15 sensors-24-04778-t015:** THD of *U_SCR_* and *I_SCR_*.

	THD of *U_SCR_* (%)			THD of *I_SCR_* (%)		
Time(s)	Phase A	Phase B	Phase C	Phase A	Phase B	Phase C
0–1	0.38	0.38	0.38	--	--	--
1–2	0.43	0.44	0.43	0.40	0.41	0.39
2–3	0.44	0.46	0.45	--	--	--

**Table 16 sensors-24-04778-t016:** Voltage deviation of *U_AC_*.

	Voltage Deviation of *U_AC_* (%)		
Time(s)	Phase A	Phase B	Phase C
0–1	1.32	1.32	1.32
1–2	0.98	0.98	0.98
2–3	0.77	0.77	0.77

**Table 17 sensors-24-04778-t017:** THD of *U_AC_* and *I_AC_*.

	THD of *U_AC_* (%)			THD of *I_AC_* (%)		
Time(s)	Phase A	Phase B	Phase C	Phase A	Phase B	Phase C
0–1	0.38	0.38	0.38	0.27	0.27	0.27
1–2	0.43	0.44	0.43	0.40	0.41	0.39
2–3	0.44	0.46	0.45	0.43	0.44	0.45

**Table 18 sensors-24-04778-t018:** THD of *I_grid_*.

	THD of *I_grid_* (%)		
Number of Converters	Phase A	Phase B	Phase C
4	2.28	2.26	2.29
6	3.64	3.58	3.60
10	5.46	5.12	5.05

## Data Availability

The data presented in this study are available on request from the corresponding author.

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
