# Peer review of "Design and Feasibility Verification of Novel AC/DC Hybrid Microgrid Structures"

_sensors, 2024, doi:10.3390/s24154778_

Round 1
Reviewer 1 Report
Comments and Suggestions for Authors
Improvements are necessary to enhance the clarity and quality of the paper. We need to make several improvements to enhance the clarity and quality of the paper:
· --- Consider these suggestions to revise your abstract and make it even more impactful: Clearly state the purpose of the research in the opening sentence; clarify the Contribution; use Precise Language; streamline Sentence Structure. The last paragraph of the abstract must include details about the study outcomes and the important results.
· --- Include a validation section that compares the current study's results with previous studies. The provided details are not enough to decide whether the model results are correct or not.
--- Study Limitations: Authors must discuss any limitations in the application of this study.
-- What are the benefits of using the suggested AC/DC grid hybrid model? Please add more details to compare this model in terms of time and cost efficiency to support the main objective of the research.
--- The paper contains a number of typographical and grammatical errors that need to be corrected.
Comments on the Quality of English Language
Grammar and Language Revision: Thoroughly revise the paper to correct typographical and grammatical errors. Conduct English language checks for improved readability.
Reviewer 2 Report
Comments and Suggestions for Authors
This paper proposes an innovative structure that allows the backup power to be accessed quickly during faults.
1. The situation that the power source fails by taking advantage of the characteristics of thyristor switches that can respond quickly is not clearly depicted.
2. The innovation and contribution of this paper is not clearly stated. The English writing of the abstract of this paper needs improving.
3. The proposed structure lacks contrast with existing structures.
4. The background of the paper is too lengthy.
Comments on the Quality of English LanguageEnglish writing needs improving.
Reviewer 3 Report
Comments and Suggestions for Authors
To improve the power supply reliability of the microgrid cluster composed of AC/DC hybrid microgrids, this paper proposes an innovative structure that allows the backup power to be accessed quickly when the power source fails by taking advantage of the characteristics of thyristor switches that can respond quickly, and gives the corresponding control strategy. In addition, this paper considers the actual situation, and respectively designs the AC/DC hybrid microgrid structures in grid-connected and islanded states, which have strong renewable energy consumption capacity, simple control, high power quality, and high practical application value.
Notes:
1. It is necessary to improve the overview of the current state of the problem. A review of each study is needed. You should not use references like [10–15], etc.
2. It is necessary to write a clear statement of the research problem
3. It is necessary to make a description of Figure 1. Topology of the novel hybrid AC/DC microgrid cluster..
4. There is no description of the mathematical model. On what basis was the study carried out?
Reviewer 4 Report
Comments and Suggestions for Authors
1. It is necessary to expand the review of hybrid ac/dc microgrids, presenting their main advantages and disadvantages, which will serve as motivation for current work.
2. It is necessary to justify the feasibility of using the proposed cluster topology. What is the need to use a thyristor switch to transfer the load from one section to another? Isn't it easier to use 2 breakers instead of K1 and K2 - this will eliminate the need for breaker B1 and a thyristor switch. What is advantages in comparison with the standard solution with Automatic Transfer Switch?
It is necessary to clarify this issue before deciding on this paper.
3. The article does not provide an explanation of the reasons for choosing 300 V voltages for DC loads.
4. Also questionable is the need to connect PV and batteries in a hybrid AC/DC microgrid to AC buses.
5. In my opinion, the term “significant load” does not correctly reflect the meaning, and “critical load” or synonyms should be used to denote consumers who must be provided with uninterrupted power supply.
6. Connecting critical loads to AC buses will eliminate the unnecessary DC/AC transformation stage, i.e. reduce costs, losses and increase reliability.
7. In Table 11 two lines in the battery parameters are mixed up.
Round 2
Reviewer 1 Report
Comments and Suggestions for Authors
The authors have made the required corrections, and the manuscript is now suitable for publication.
Author Response
Thank you very much for your prompt review of our manuscript. It is thanks to your valuable comments that we have been able to continuously improve and refine our work. We sincerely hope to have the opportunity to continue receiving your guidance in the future.
Reviewer 4 Report
Comments and Suggestions for Authors
Thank you for you response. There are some questions that still should be reflected in the text:
1. The only significant advantage of a thyristor switch is its speed. For ordinary consumers, and even more so for microgrids operating in island mode, such speed is not required. Thus, the use of this technology is advisable only if there are consumers for whom an interruption in power supply during the operation of the automatic transfer switch is critical. Are there such consumers in microgrids?
2. In real operation, the reliability of thyristor switches for high current load often does not correspond to the declared characteristics, which in reality can lead to a decrease in the level of reliability of power supply.
